# Insights into tuberculosis burden in Karachi, Pakistan: A concurrent adult tuberculosis prevalence and child *Mycobacterium tuberculosis* infection survey

Palwasha Y. Khan[1,2,3]*, Mohammed Shariq Paracha[4], Chris Grundy[5],
Falak Madhani[4,6], Saadia Saeed[7], Lamis Maniar[8], Maqboola Dojki[9], Liesl Page-Shipp[3],
Nazia Khursheed[9], Waleed Rabbani[10], Najam Riaz[3], Saira Khowaja[3], Owais Hussain[11],
Rabia Maniar[3], Uzma Khan[3], Salman Khan[12], Syed S. H. Kazmi[12], Ali A. Dahri[12],
Abdul Ghafoor[13], Sabira Tahseen[14], Ali Habib[15], James J. Lewis[16], Katharina Kranzer[1],
Rashida A. Ferrand[1,17], Katherine L. Fielding[5], Aamir J. Khan[3]

1 Department of Clinical Research, London School of Hygiene & Tropical Medicine, London School of Hygiene & Tropical Medicine, London, United Kingdom, 2 Africa Health Research Institute, Durban, South Africa, 3 Interactive Research and Development Global, Singapore, Singapore, 4 Aga Khan Health Services, Karachi, Pakistan, 5 Department of Infectious Disease Epidemiology, London School of Hygiene & Tropical Medicine, London, United Kingdom, 6 Faculty of Arts and Sciences, Aga Khan University, Karachi, Pakistan, 7 The World Bank, Karachi, Pakistan, 8 School of Arts, Humanities and Social Sciences, Habib University, Karachi, Pakistan, 9 Indus Hospital Network, Karachi, Pakistan, 10 Interactive Research and Development, Karachi, Pakistan, 11 Institute of Economics and Technology, Karachi, Pakistan, 12 TB Control Programme, Communicable Disease Control-CDC Sindh, Hyderabad, Pakistan, 13 MDR-TB Department, National TB Control Programme, Islamabad, Pakistan, 14 National TB Reference Laboratory, National TB Control Programme, Islamabad, Pakistan, 15 Interactive Health Solutions, Karachi, Pakistan, 16 Cardiff University, Y Lab–the Public Services Innovation Lab for Wales, School of Social Sciences, Cardiff, Wales, 17 Department of Paediatrics, Aga Khan University, Karachi, Pakistan

* palwasha.khan@lshtm.ac.uk

**Data Availability Statement:** The data that support the findings of this study are not publicly available

## Abstract

Pakistan is one of the five highest tuberculosis burden countries globally. We estimated prevalence of adult bacteriologically confirmed pulmonary tuberculosis and annual risk of *Mycobacterium tuberculosis (M. tuberculosis)* infection in children aged 2–4 years in Karachi, Pakistan. The survey design enabled exploration of tuberculosis burden by whether the population had previously been exposed to widespread tuberculosis active case-finding (ACF) activities or not. We conducted a concurrent adult pulmonary tuberculosis prevalence survey and a child *M. tuberculosis* infection survey using interferon gamma release assays in four districts (Korangi, South, West and Central). A cluster-based unequal probability random sampling method was employed with the *a priori* plan to oversample Korangi district which had been the focus of tuberculosis ACF activities since 2011. We defined Korangi district as the 'prior ACF' zone and remaining districts as the 'no prior ACF' zone. Between March 2018 and May 2019, 34,962 adults (78·5% of those eligible) and 1,505 children (59·9%) participated. Overall estimated prevalence of bacteriologically confirmed pulmonary tuberculosis was 387 cases per 100,000 population (95% CI 276–498) with a prevalence of 421 cases [95% CI 276–567] per 100,000 in the 'no prior ACF' and 279 cases [95% CI 155–403] per 100,000 in the 'prior ACF' zone. We estimated the annual risk of *M. tuberculosis*

due to ethical restrictions. Ethical clearance for specific data requests may be sought in collaboration with the corresponding author from the Ethics Committee of the London School of Hygiene & Tropical Medicine. Email address for LSHTM Ethics Committee is ethics@lshtm.ac.uk.

**Funding:** This study was funded in part by the Global Fund Against Tuberculosis, AIDS and Malaria (GFATM) (Grant ID: PAK-T-TIH) as part of larger R9 grant awarded to the Indus Hospital Network and in part by Interactive Research and Development (IRD). GFATM had no role in the study design, data collection, data analysis and interpretation, decision to publish and preparation of the manuscript. IRD part funded the study team which conducted the data collection led by PYK. IRD had no role in the data analysis, interpretation or decision to publish.

**Competing interests:** The authors have declared that no competing interests exist.

infection in children to be 1·1% (95% CI 0·7–1·5) in the 'no prior ACF' zone and 0·6% (95% CI 0·3–1·1) in the 'prior ACF' zone. We observed consistent differences in the population distribution of tuberculosis between the 'prior ACF' and 'no prior' ACF zones with a trend towards lower estimates of burden and *M. tuberculosis* transmission in the 'prior ACF' zone. A plausible explanation is that intensive ACF activities that have been ongoing in Korangi district for the preceding years have noticeably reduced the burden of tuberculosis and transmission.

## Introduction

The 2022 World Health Organization (WHO) Global TB report was the first time in many years that an increase in estimated tuberculosis burden compared to the previous year was officially announced with over 10.6 million people falling ill and 1.6 million dying from tuberculosis. The deleterious impact of coronavirus disease 2019 (COVID-19) pandemic on global tuberculosis care and prevention is likely to be felt for many years to come [1]. However, from a public health perspective, the COVID-19 pandemic has helped to highlight the importance of methodologically rigorous descriptive epidemiology in guiding the response to serious public health crises [2]. By acquiring valuable insights of tuberculosis burden and *Mycobacterium tuberculosis* (*M. tuberculosis*) infection burden in sub-populations over space and time, the tuberculosis community too would be in a position to understand the distribution of disease and thus design, implement and test care and prevention interventions tailored to the context that would likely be more effective in reducing *M. tuberculosis* transmission than the current 'one-size fits all' approach.

The transmission dynamics of many, if not all, infectious pathogens that transmit from human to human are dependent on the ecology (social and demographic characteristics and contact patterns) as well as the historical experiences of the human host population (e.g., previous infection, vaccination and interventions to reduce transmission) [3]. However understanding what determines current prevalence and incidence rates and predicting future burden is complex for tuberculosis [4,5]. Interpretation of changes in adult pulmonary disease prevalence at a population-level needs to be supported by simultaneously undertaking a survey which measures a parameter of *M. tuberculosis* transmission at a population-level, such as risk of *M. tuberculosis* infection [6,7]. However over the last several decades, these have rarely been conducted concurrently to estimate burden [6].

Tuberculosis active case finding (ACF) is one intervention that may help to reduce onward transmission of *M. tuberculosis* by finding and linking people affected by tuberculosis care early in their disease course and thereby reduce duration of infectiousness [8], but these activities are resource-intensive and the population-level impact of ACF remains uncertain [9,10]. Over the last decade, ACF activities have been rolled out in many settings but to date there are very little data on the potential population-level impact on transmission outside of trial settings [11].

In this manuscript, we present findings from our survey undertaken in Karachi, the largest city of Pakistan—one of the five highest tuberculosis burden countries globally—in 2018 to 2019 to estimate the burden of adult pulmonary tuberculosis and the annual risk of *M. tuberculosis* infection in children aged 2–4 years. ACF has been undertaken at scale in certain parts of the city over the preceding seven years. The design of the survey allowed us to examine tuberculosis prevalence and transmission by whether the population had been previously exposed to widespread tuberculosis ACF activities or not.

## Materials and methods

### Study design and setting

A concurrent adult pulmonary tuberculosis and child *M. tuberculosis* infection prevalence survey using an unequal probability cluster-based random sampling approach was undertaken in Karachi (estimated population of 16 million) [12] from March 2018 to May 2019. Karachi is the largest city in Pakistan, a country which had the highest rate of urbanisation in South Asia in 2018 [13].

The rationale for this survey was to primarily provide a baseline measure of the burden of infectious tuberculosis (bacteriologically confirmed adult pulmonary tuberculosis) in Karachi in 2018 to 2019. This was at the start of a fully scaled up implementation of a tuberculosis elimination strategy named Zero TB Karachi initiative: a community-based comprehensive 'search-treat-prevent' approach of searching actively for tuberculosis cases, treating effectively and preventing exposure.[14] Due to logistical constraints, namely travel time and number of teams, the survey was restricted to four districts (Korangi, South, Central and West) which comprised approximately 65% of the total population of Karachi. All included districts are each made up of a combination of formal and informal settlements (slums) known locally as *katchi abadis*. The survey covers both the formal and informal settlements to ensure a high-quality probability sample of the population. For further details see published protocol [15].

Prior to the Zero TB Karachi initiative, Interactive Research and Development (IRD), a not-for-profit organisation, implemented several grants aimed at increasing tuberculosis case notifications in the private sector over the period of 2010–2017 in Korangi district. This district was the designated focus for implementation as it was identified as one of the most disadvantaged in Karachi with limited access to healthcare. The first of these grants funded an extensive ACF campaign in 2010–2011, which resulted in a doubling of tuberculosis notification rates in Korangi district compared to an adjacent control area [16]. Varying intensity of ACF has continued in Korangi district since 2011. In view of the historical context of ACF activities in Korangi district, an *a priori* decision was made to oversample this district. We defined Korangi district as the 'prior ACF' zone with Karachi South, Central and West districts defined as the 'no prior ACF' zone although some ACF activities had started to be undertaken in districts outside of Korangi from 2016 onwards. S1 Fig shows the trend over time of case notification rate per 100,000 population of new adult bacteriologically confirmed pulmonary tuberculosis by ACF zone from 2007 to 2017.

The design of the survey therefore allowed us to explore tuberculosis burden by whether the population had previously been exposed to widespread tuberculosis ACF activities or not.

### Participants

Neighbourhood blocks (defined as geographically adjacent households with approximately 200 individuals aged ≥15 years) were identified and randomly selected using ArcGIS software (Environmental Systems Research Institute, version 10.5, Redlands, CA, USA) using probability proportional-to-population size of each *tehsil (*an administrative district subdivision) within each district in the 'no prior ACF' zone (Karachi South, Central and West districts) with oversampling in the 'prior ACF' zone (Korangi district) as defined in the protocol [15]. A set number of neighbourhood blocks from *katchi abadi* (informal settlements) within each district were sampled to be consistent with the overall proportion of population residing in *katchi abadis* at the district-level. All adults aged ≥15 years who were resident within households in the neighbourhood block and gave verbal consent were eligible for inclusion in the tuberculosis prevalence survey. All children aged 2 to 4 years who were resident in the neighbourhood

block whose guardians gave verbal consent were eligible for inclusion in the *M. tuberculosis* infection survey. Residency was defined as having slept in the household the night before enumeration of the neighbourhood block.

## Study procedures

**Adult tuberculosis prevalence survey.** The procedures of the adult tuberculosis prevalence survey were based on the WHO methodology for tuberculosis prevalence surveys [17]. Participants underwent a questionnaire including information on previous and current tuberculosis treatment, tuberculosis symptoms, history of smoking and diabetes, and were referred to the x-ray van stationed in the neighbourhood for a digital chest x-ray (CXR). Computer-aided software for reading radiological signs of tuberculosis, CAD4TB version 5 (Diagnostic Image Analysis Group, Radboud University Medical Center, Nijmegen, the Netherlands) was used to evaluate the digital CXR. A cut off CAD4TB score of 65 was used in the survey to identify an 'abnormal' CXR based on analysis of local programmatic data examining the association of CAD4TB score and prevalence of Xpert-positivity [18]. If either the tuberculosis symptom screen was positive or CAD4TB score $\geq$ 65, individuals were asked to submit two sputum samples, an instructed spot sputum sample on the same day and an early morning sputum. All early morning samples were tested with Xpert MTB/RIF Ultra (Xpert Ultra: Cepheid, Sunnyvale, CA, USA) and liquid culture (BACTEC Mycobacterial Growth Indicator Tube (MGIT) 960 [BD])) and solid media (Lowenstein Jensen (LJ) [BD]) and on spot samples were investigated with liquid and solid media cultures aiming for 2 cultures from 2 separate sputum samples and an Xpert Ultra on all who screened positive based on symptoms and/or CAD4TB score.

Individuals who were already on tuberculosis treatment at the time of screening were asked to submit two (on the spot and early morning) sputum samples, irrespective of symptom screen and digital CXR. Individuals with physical disabilities that prevented them from being able to mobilise to the x-ray van and pregnant women underwent a symptom screen only and were also asked to submit two sputum samples irrespective of symptoms. All study participants who were microbiologically confirmed including Xpert Ultra 'trace positive' only on any samples were referred to the nearest health facility for clinical review and to start tuberculosis treatment if appropriate.

**Child M. tuberculosis infection survey using interferon-gamma release assays (IGRA).** Guardians of children aged 2 to 4 years old completed a brief verbal questionnaire about their children including a symptom screen (cough, fever, weight loss, failure to thrive and decreased playfulness), history of tuberculosis treatment, and history of any known tuberculosis contact within the last two years. A 4ml sample of blood was collected irrespective of child symptom screen status and transported to the Indus Hospital central laboratory within 6 hours of venepuncture for IGRA testing using the QuantiFERON-TB Gold Plus assay. Children with positive IGRA results were referred to the closest of one of three hospitals, which provide specialist childhood tuberculosis care in Karachi, where the child was assessed and evaluated for tuberculosis, including a full diagnostic work-up by a trained paediatrician. Once tuberculosis had been excluded, children were started on a 6-month course of isoniazid preventive therapy and followed-up for duration of treatment by the childhood tuberculosis programme.

The data collection tool was a custom-built application for Android tablets based on the OpenMRS platform [19] and co-developed by the principal investigator (PYK) and the software development team, Interactive Health Solutions (IHS: AH, OH). The laboratory results of all tests conducted were entered into the associated OpenMRS software installed in the

laboratory using the unique study identifier and specimen number which allowed linkage of the field and laboratory data within the study database.

## Statistical methods

**Sample size and sampling design.** The overall target sample size for the tuberculosis prevalence survey was 44,000 adults aged ≥15 years. The sample size was substantially larger than the sample size needed to estimate the prevalence of adult pulmonary tuberculosis in Karachi in 2018–2019 (n = 21,668 based on the assumptions of a prevalence of bacteriologically confirmed pulmonary tuberculosis of 400 per 100,000, relative precision of 25%, a design effect of 1·128 and a participation rate of 75%). The reason for this inflation from approximately 22,000 to 44,000 was to provide enough power to detect a difference in a planned 'before and after' comparison as part of an impact evaluation of Zero TB Karachi initiative later down the line [15].

**Statistical analysis.** *Adult survey case definition*: we included all probable and definite bacteriologically confirmed pulmonary tuberculosis (defined as per protocol) [15], i.e. any form of microbiological confirmation (including Xpert Ultra 'trace positive' only) on at least one specimen, for the purposes of estimating the prevalence of adult pulmonary tuberculosis. Xpert Ultra 'trace positive' only with a previous history of tuberculosis were excluded from analysis. If a participant had no symptoms and the CAD4TB score<65 then Xpert Ultra and culture results were assumed to be negative.

We followed the WHO recommended best practice analytical methods to estimate the prevalence of bacteriologically confirmed pulmonary tuberculosis [20]. These methods account for the cluster sampling design, non-participation and missing data. We present cluster-level estimates based on the geometric mean at the *tehsil* level (n = 13 *tehsils*) using sampling weights at district-level to account for the unequal probability of selection (for example, individuals resident in Korangi district were twice as likely to be selected compared to the individuals resident in the three other districts) and two individual-level model-based estimates which use logistic regression taking into account the clustered design using robust standard errors (i) complete case analysis and (ii) with multiple imputation for individuals eligible for sputum submission and inverse probability weighting for all survey participants. As a sensitivity analysis we repeated the above analyses excluding all Xpert Ultra 'trace positive' culture-negative cases.

*Child survey case definition*: *M. tuberculosis* infection in children was defined as having a positive IGRA test as per definition recommended by the manufacturer.

The prevalence of *M. tuberculosis* infection in 2 to 4 year-olds allows the derivation of an average annual risk of *M. tuberculosis* infection risk (ARTI), under the assumption of no change in risk over calendar time of the lifetime of the cohort tested [21], i.e. no change in risk over the last 3 years.

The annual risk of infection (ARTI) was estimated as:

$$\text{ARTI} = 1 - (1 - \text{P})^{1/a}$$

where *P* is prevalence of *M. tuberculosis* infection, excluding indeterminates, and *a* is the mean age at which the observed prevalence is estimated. Two methods were used to characterise the ARTI: (i) complete case data with a non-parametric bootstrap procedure with resampling was performed with 20,000 replicates to generate a 95% uncertainty interval using the percentile method. A non-parametric bootstrapping method was used as there is no clear consensus of the underlying theoretical distribution of the ARTI; (ii) imputation for missing IGRA results and inverse probability weighting for all IGRA survey participants.

All models are described in (see S1 Text). To estimate the population count, we used WorldPop data for Pakistan from 2017 which estimates number of people per pixel using a Random Forest-based dasymetric redistribution mapping approach [22]. All statistical analyses were conducted in Stata version 17 (Stata Corporation, College Station, TX, USA) and R version 4.2.2 (R Foundation for Statistical Computing, Vienna, Austria).

**Ethics approval.** The study was approved by the ethical review committee of Interactive Research Development (IRD) [IRD reference number: IRD_IRB_2017_04_002] and the London School of Hygiene & Tropical Medicine (LSHTM) [LSHTM ethics reference number: 12063]. Verbal informed consent was obtained from each eligible participant aged ≥15 years before enrolment into the study after a thorough explanation of the risks and benefits of participation. Consent for minors aged 2 to 4 years was sought from a parent or guardian. We specifically applied for a waiver with respect to obtaining written informed consent from both ethics review committees to minimise selection bias in this setting where a large proportion of the population are illiterate and are reluctant to provide fingerprints due to concerns about personal security.

**Role of the funding source.** This study was funded by the Global Fund Against Tuberculosis, AIDS and Malaria (GFATM) and IRD Global. GFATM had no role in the study design, data collection, data analysis and interpretation, decision to publish and preparation of the manuscript.

## Results

Between March 2018 and May 2019, a median of 39 and 42 buildings per neighbourhood blocks were enumerated, in the no prior ACF zone and ACF zone respectively, with 86% and 87% of buildings accessed.

In total, 64,099 people were enumerated from 230 randomly selected neighbourhood blocks (cluster size with median 175 individuals aged ≥ 15 years) of whom 44,565 adults aged ≥ 15 years and 2,511 children aged 2 to 4 years were eligible to partake in the survey. Of those eligible, 34,964 adults (78·5%) and 1,505 children (59·9%) participated in the adult pulmonary tuberculosis survey and child *M. tuberculosis* infection survey respectively. Participation percentage in the adult survey was higher in women than in men, 90.1% [19,898/22,073] versus 67.0% [15,066/22,492]. S2A and S2B Fig in the supplementary material include the study flowcharts which summarise the numbers of individuals at each stage of the adult survey and the child *M. tuberculosis* infection survey.

### Adult pulmonary tuberculosis survey

**Clinical characteristics.** Table 1 summarises the characteristics of the study participants by zone and sex. The mean age of study participants were similar across all groups. The proportion reporting any cough were similar across all groups ranging from 5% to 9% and only approximately 1% of all study participants reported having a cough longer than two weeks duration. The proportion on tuberculosis treatment at time of survey and also the proportion reporting known TB contact in past two years were similar overall by ACF zone. Median CAD4TB scores were highest for those aged >45 years (Fig 1). The shape of the distributions by zone were similar but the median score in each sex-age category in the prior ACF stratum was lower than the corresponding value in the no prior ACF.

**Sputum submission and microbiological findings.** Overall 10·8% (3,786/34,964) of all study participants who participated were eligible for sputum submission. The proportion of those who submitted at least one sputum sample was lower in the no prior ACF group compared to the prior ACF group (63·9% versus 73·0%). Only 29/93 (31%) of those identified with

**Table 1. Characteristics of participants and microbiological outcomes of the adult pulmonary tuberculosis survey stratified by zone and sex.**

| | No prior ACF | | | Prior ACF | | |
|---|---|---|---|---|---|---|
| | **Male** | **Female** | **Overall** | **Male** | **Female** | **Overall** |
| **Survey participation** | | | | | | |
| Eligible | 11,191 | 11,106 | 22,297 | 11,301 | 10,967 | 22,268 |
| Consented to participate | 6,994 | 9,932 | 16,926 | 8,072 | 9,966 | 18,038 |
| % participation | 62·5% | 89·4% | 75·9% | 71·4% | 90·9% | 81·0% |
| **Baseline characteristics (n = 34,964)** | | | | | | |
| Age (years), mean (sd) | 33·6 (15·6) | 33·2 (14·4) | 33·4 (14·9) | 33·5 (15·2) | 33·3 (14·3) | 33·4 (14·7) |
| Symptom status | | | | | | |
| Any cough (n = 34,896) | 600 (8·6%) | 500 (5·1%) | 1100 (6·5%) | 689 (8·6%) | 661 (6·6%) | 1350 (7·5%) |
| Cough ≥ 2 weeks (n = 34,896) | 75 (1·07%) | 76 (0·77%) | 151 (0·89%) | 61 (0·76%) | 77 (0·77%) | 139 (0·77%) |
| Any TB symptom^ | 104 (1·49%) | 128 (1·29%) | 232 (1·37%) | 100 (1·24%) | 133 (1·33%) | 233 (1·29%) |
| Sought medical care if symptoms (n = 451) | 17 (16·5%) | 41 (35·7%) | 58 (26·6%) | 30 (30·0%) | 47 (35·3%) | 77 (33·1%) |
| On TB treatment (n = 34,875) | 16 (0·23%) | 14 (0·14%) | 30 (0·18%) | 23 (0·29%) | 22 (0·22%) | 45 (0·25%) |
| Previous history of TB (n = 34,831) | 113 (1·62%) | 235 (2·37%) | 348 (2·06%) | 94 (1·17%) | 197 (1·98%) | 291 (1·62%) |
| Known TB contact within last 2 years (n = 34,807) | 180 (2·58%) | 364 (3·68%) | 544 (3·23%) | 175 (2·18%) | 308 (3·11%) | 483 (2·69%) |
| Self-reported having diabetes (n = 34,332) | 162 (2·35%) | 397 (4·06%) | 559 (3·36%) | 231 (2·93%) | 421 (4·30%) | 652 (3·69%) |
| Smoking status | | | | | | |
| Ex-smoker | 57 (0·81%) | 9 (0·09%) | 66 (0·39%) | 52 (0·64%) | 10 (0·10%) | 62 (0·34%) |
| Currently smoking | 781 (11·2%) | 177 (1·78%) | 958 (5·7%) | 1057 (13·1%) | 44 (0·44%) | 1101 (6·1%) |
| CXR in last 3 months (n = 31,175) | 59 (0·90%) | 63 (0·76%) | 122 (0·82%) | 80 (1·05%) | 87 (1·01%) | 167 (1·02%) |
| Self-reported pregnancy (n = 19,801) | - | 280 (2·83%) | - | - | 281 (2·84%) | - |
| Self-reported disability | 15 (0·21%) | 27 (0·27%) | 42 (0·25%) | 17 (0·21%) | 24 (0·24%) | 41 (0·23%) |
| Digital chest x-ray (n = 30,892) | | | | | | |
| Number x-rayed (% of consented) | 6491 (92·8%) | 8260 (83·2%) | 14751 (87·2%) | 7573 (93·8%) | 8568 (86·0%) | 16141 (89·5%) |
| Median CAD4TB score (IQR) | 46 (35–56) | 47 (37–55) | 47 (36–56) | 44 (33–55) | 46 (36–54) | 45 (35–55) |
| CAD4TB ≥ 65 (% x-rayed) | 741 (11·4%) | 678 (8·2%) | 1419 (9·6%) | 729 (9·6%) | 658 (7·7%) | 1387 (8·6%) |
| **Sputum submission eligibility (n = 34,946)** | | | | | | |
| Unable to establish eligibility due to missed CXR (n = 3,111) | 429 (6·2%) | 1286 (13·0%) | 1715 (10·1%) | 397 (4·9%) | 999 (10·0%) | 1396 (7·7%) |
| Not eligible for sputum submission (n = 28,018) | 5738 (82·0) | 7544 (76·0%) | 13282 (78·5%) | 6849 (84·9%) | 7887 (79·2%) | 14736 (81·7%) |
| Eligible for sputum submission* (n = 3,835) | 827 (11·8%) | 1102 (11·0%) | 1929 (11·4%) | 826 (10·2%) | 1080 (10·8%) | 1906 (10·6%) |
| Unable to expectorate (n = 439) | 97 (11·7%) | 136 (12·3%) | 233 (12·1%) | 100 (12·1%) | 106 (9·8%) | 206 (10·8%) |
| Submitted at least one sample (n = 2,599) | 526 (64·3%) | 699 (64·2%) | 1225 (63·9%) | 584 (71·2%) | 790 (74·4%) | 1374 (73·0%) |
| *All samples rejected by lab* | *11 (1·35%)* | *8 (0·74%)* | *19 (1·00%)* | *10 (1·24%)* | *5 (0·47%)* | *15 (0·81%)* |
| **Microbiological status if at least one sputum sample tested (n = 2,565)** | | | | | | |
| Valid result for Xpert Ultra (% of eligible for sputum submission) | 507 (61·3%) | 681 (61·8%) | 1188 (61·6%) | 562 (68·0%) | 766 (70·9%) | 1328 (69·7%) |
| Valid result for culture (% of eligible for sputum submission) | 526 (63·6%) | 698 (63·3%) | 1224 (63·5%) | 584 (70·7%) | 790 (73·1%) | 1374 (72·0%) |
| Xpert Ultra & culture negative | 485 | 668 | 1153 | 557 | 762 | 1319 |
| **Any form of microbiological confirmation** (% of at least one sputum submitted) | 30 (5·7%) | 23 (3·3%) | 53 (4·3%) | 17 (2·9%) | 23 (2·9%) | 40 (2·9%) |
| Xpert Ultra-negative & culture positive | 2/30 (6·6%) | 2/23 (8·7%) | 4/53 (7·5%) | 2/17 (11·8%) | 6/23 (26·0%) | 8/40 (20·0%) |
| Xpert Ultra 'trace positive' only ** | 13/30 (43·3%) | 13/23 (56·5%) | 26/53 (49·0%) | 1/17 (5·9%) | 10/23 (43·5%) | 11/40 (27·5%) |
| Xpert Ultra-positive & culture negative | 5/30 (16·7%) | 0/23 (0%) | 5/53 (9·4%) | 4/17 (23·5%) | 4/23 (17·4%) | 8/40 (20·0%) |
| Xpert Ultra-positive & culture positive | 10/30 (33·3%) | 8/23 (34·8%) | 18/53 (34·0%) | 10/17 (58·8%) | 3/23 (13·0%) | 13/40 (32·5%) |
| **Follow-up and treatment status if microbiologically positive sputum sample (n = 93)** | | | | | | |
| Unable to contact | 6 | 6 | 12 (22·7%) | 7 | 4 | 11 (27·5%) |
| Contacted and refused follow-up | 10 | 12 | 22 (41·5%) | 3 | 5 | 8 (20·0%) |

*(Continued)*

**Table 1.** (Continued)

| | No prior ACF | | | Prior ACF | | |
|---|---|---|---|---|---|---|
| | **Male** | **Female** | **Overall** | **Male** | **Female** | **Overall** |
| Reviewed clinically and TB excluded | 5 | 2 | 7 (13·2%) | 0 | 9 | 9 (22·5%) |
| Started treatment | 9 | 3 | 12 (22·6%) | 7 | 5 | 12 (30·0%) |

^ Cough ≥ 2 weeks, weight loss, fever, night sweats.

* Eligibility for sputum included all those who screened positive on symptoms or CAD4TB≥65 or were disabled or pregnant (no chest x-ray) or on TB treatment.

** none culture-positive.

sd standard deviation; TB tuberculosis; Ultra Xpert MTB/RIF Ultra; culture with liquid and solid media; IQR interquartile range, CAD4TB computer-aided software for reading radiological signs of tuberculosis; CXR chest x-ray; ACF active case finding.

*M. tuberculosis* in their sputum reported any cough or any tuberculosis-related symptom. We also identified five pregnant women with evidence of *M. tuberculosis* in their sputum all of whom reported no symptoms.

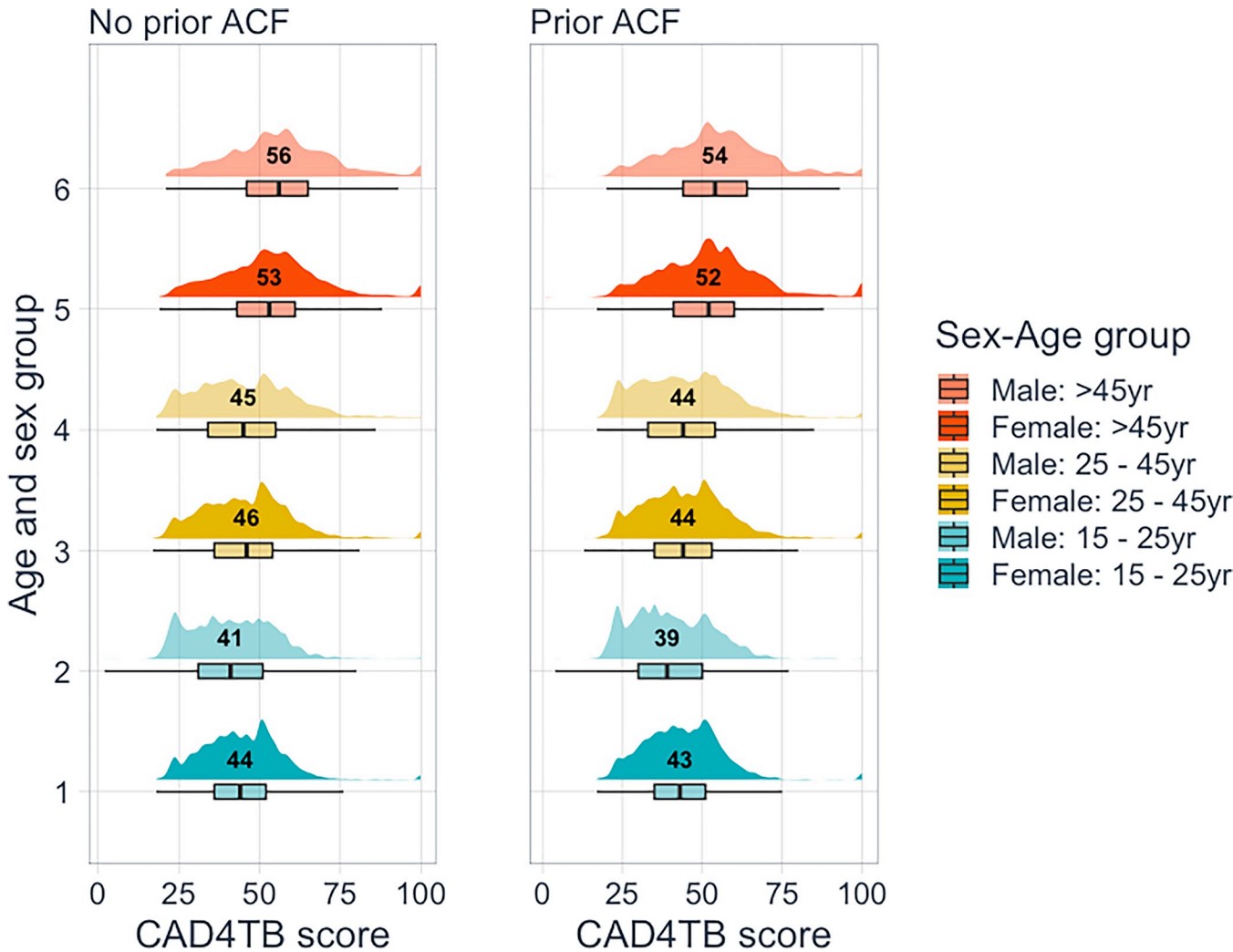

**Fig 1. Distribution of CAD4TB score by zone, age and sex (median per group displayed).**

Overall 40% (37/93) of all microbiologically confirmed cases were Xpert Ultra 'trace positive' only; 3/37 were in pregnant women (see S1 Table for description of these cases). The proportion of microbiologically confirmed cases that were Xpert Ultra-positive and culture-positive were similar across the two zones (34% in no prior ACF versus 33% in prior ACF zone). S4 Fig illustrates the distribution of CAD4TB score by microbiological status.

Clinical characteristics, sputum submission and microbiological findings stratified by district are shown in S2 Table. No striking heterogeneity across the 3 districts of the no prior ACF zone were observed.

**Prevalence estimates by ACF zone and sex.** Of the 93 study participants identified with *M. tuberculosis* in their sputum, 4 with Ultra 'trace positive' only and a previous history of tuberculosis were excluded from prevalence estimate analyses. Results for the prevalence estimates of adult pulmonary tuberculosis are summarised in Table 2. For the cluster-level analysis including sampling weights, the overall prevalence estimate was 275 per 100,000 population (95% CI: 204–407) with a trend to a lower estimate in the prior ACF zone (183 per 100,000: 95% CI 94–353) compared to the no prior ACF zone (288 per 100,000: 95% CI 204–407). The trend towards a lower estimate in the prior ACF compared to no prior ACF was seen irrespective of the method used. Of note, differences in prevalence estimates by sex were observed across zones. The prevalence estimates were higher in men compared to women in the 'no prior ACF' irrespective of method used and were similar in men and women in the 'prior ACF' zone. The sensitivity analysis (S3 Table) which excluded all study participants with Ultra 'trace positive' culture-negative resulted in similar prevalence estimates in the two zones.

**Prevalence estimates by ACF zone and katchi abadi status.** S4 Table summarises the estimates by *katchi abadi* status and zone. The prevalence estimates in the prior ACF in *katchi abadi*s were markedly lower than the estimate for the no prior ACF zone. However the 95% confidence intervals are wide reflecting the small number of microbiologically confirmed cases detected.

**Linkage to care and treatment initiation.** Of all the study participants confirmed to have presence of *M. tuberculosis* in their sputum, 57% (53/93) either refused further contact or were

**Table 2. Adult pulmonary tuberculosis prevalence estimates stratified by zone and sex (per 100,000).**

| | Overall point prevalence (95% CI) | No prior ACF | | | Prior ACF | | |
|---|---|---|---|---|---|---|---|
| | | Male | Female | Overall | Male | Female | Overall |
| Cluster-level | | | | | | | |
| N = 13* (n = 9; n = 4) | 275 (191–398) | 501 (300–837) | 205 (123–342) | 345 (235–506) | 180 (71–459) | 174 (69–443) | 183 (98–341) |
| Including sampling weights | 288 (204–407) | 480 (323–714) | 189 (91–391) | 306 (209–449) | 180 (61–533) | 174 (46–659) | 183 (94–353) |
| Individual level | | | | | | | |
| Complete case N = 30,579 (n = 14,486; n = 16,093) | 291 (209–374) | 463 (328–599) | 267 (101–433) | 352 (236–467) | 230 (133–324) | 242 (101–383) | 236 (131–341) |
| IPW/MI N = 31,849** (n = 15,209; n = 16,640) | 387 (276–498) | 530 (346–714) | 304 (111–498) | 421 (276–567) | 283 (152–414) | 274 (138–410) | 279 (155–403) |

IPW/MI inverse probability weighting and multiple imputation using chained equations.

ACF active case finding CI confidence interval.

*Clusters relate to the 13 tehsils (9 in the no prior ACF zone and 4 in the prior ACF zone).

**N = 30,579 + 1270 (multiple imputation of subset of survey participants who were eligible for sputum examination but for whom Xpert Ultra and/or culture results were missing).

unable to be traced; 17% (16/93) were reviewed clinically and active tuberculosis was excluded and 26% (24/93) started treatment. Of those that were reviewed and tuberculosis excluded, 75% (12/16) were Xpert Ultra 'trace positive' only (median CAD4TB score of 71·5) and 2/16 had had a previous episode of tuberculosis treated more than 4 years prior.

**Child M. tuberculosis infection (IGRA) survey.** The individual-level characteristics of the children included in the concurrent IGRA survey with neighbourhood block-level characteristics stratified by ACF zone are shown in Table 3. The participation and baseline characteristics of the children included were similar across zones. Although average population count within 100 metres of neighbourhood blocks was higher in the no prior ACF zone compared to the prior ACF zone, as seen in S3 Fig. In addition, the proportion of children who have had

**Table 3. Characteristics of children aged 2–4 years, neighbourhood block and IGRA testing.**

| | No prior ACF | Prior ACF |
|---|---|---|
| | N (col %) | N (col %) |
| **Survey participation (n = 2,511 eligible)** | | |
| Eligible | 1395 | 1116 |
| Guardian consented to symptom screen | 861 (62%) | 644 (58%) |
| Guardian/parent refusal | 261 (18%) | 238 (21%) |
| Missed* | 273 (20%) | 234 (21%) |
| Age in years, mean (sd) | 3·1 (0·8) | 3·1 (0·8) |
| Female | 440 (49%) | 346 (46%) |
| Median number of adults resident in household [IQR] | 4 (2–7) | 4 (2–6) |
| **Questionnaire (n = 1505)** | | |
| Symptom status reported by guardian | | |
| Cough (n = 1502) | 169 (20%) | 152 (24%) |
| Fever (n = 1503) | 86 (10%) | 80 (12%) |
| Weight loss (n = 1483) | 100 (12%) | 78 (12%) |
| Failure to thrive (n = 1499) | 80 (9%) | 60 (9%) |
| Decreased playfulness (n = 1500) | 47 (6%) | 36 (6%) |
| At least one symptom | 252 (29%) | 219 (34%) |
| Previous history of TB (n = 1502) | 4 (0·5%) | 4 (0·6%) |
| Known TB contact in last 2 years (n = 1499) | 21 (2·4%) | 30 (4·7%) |
| Neighbourhood block-level characteristics | | |
| Median CAD4TB score [IQR] | 46 [43–49] | 45 [43–47] |
| Average population count within 100m, median [IQR] | 288 [96–394] | 194 [143–255] |
| Proportion eligible in *katchi abadi* | 113/1395 (8·1%) | 107/1116 (9·6%) |
| Proportion participated in *katchi abadi* | 59/861 (6·9%) | 70/644 (10·9%) |
| **IGRA testing (n = 1,505)** | | |
| No blood sample collection (n = 78) | | |
| Guardian/parent refused blood test | 16 | 15 |
| Child not at home /phlebotomist not available | 6 | 11 |
| Phlebotomist unable to bleed | 11 | 19 |
| **IGRA results (n = 1,427)** | 828 (59%) | 599 (54%) |
| Indeterminate | 24 (2·9%) | 11 (1·8%) |
| Negative | 777 (94%) | 576 (96%) |
| Positive | 27 (3·3%) | 12 (2·0%) |

*Mother/aunt/grandmother/carer felt unable to give to consent without father present

sd standard deviation; IGRA Interferon-gamma release assay, TB tuberculosis; IQR interquartile range, CAD4TB computer-aided software for reading radiological signs of tuberculosis; CXR chest x-ray; ACF active case finding.

Table 4. Child IGRA prevalence and annual risk of *M.tuberculosis* infection estimates.

| | Complete case (n = 1,392) | | | Point prevalence estimates using IPW/MI model (95% CI)** | | |
|---|---|---|---|---|---|---|
| | **Overall** | **No prior ACF** | **Prior ACF** | **Overall** | **No prior ACF** | **Prior ACF** |
| *M.tuberculosis infection* prevalence* | 39/1392 (2·8%) | 27/804 (3·4%) | 12/588 (2·0%) | 3.0% (1·8–4·1) | 3·5% (2·3–4·8) | 2·3% (0·9–3·6) |
| Mean age | 3·08 | 3·09 | 3·05 | 3.04 | 3.05 | 3.03 |
| ARTI (95% confidence intervals)^ | 0·9% (0·6–1·2)^ | 1·1% (0·7–1·5)^ | 0·6% (0·3–1·1)^ | 1·0% (0·6–1·4)$ | 1·2% (0·8–1·6)$ | 0·8% (0·3–1·2)$ |

IGRA Interferon-gamma release assay, ACF active case finding; ARTI annual risk of *M.tuberculosis* infection.

* *M.tuberculosis* infection prevalence as inferred from prevalence of IGRA-positivity (excluding indeterminates).

^ non-parametric bootstrap sampling (n = 20,000) used to generate uncertainty intervals around ARTI estimates.

$ estimates from IPW/MI model used to generate 95% confidence intervals.

** N = 2,511 imputed IGRA results for n = 1,119 with IPW to adjust for survey participation.

contact with a tuberculosis patient in last 2 years was higher in prior ACF zone than no prior ACF zone, 4·7% versus 2·4%.

**Prevalence estimates and annual risk of M. tuberculosis infection (ARTI) by ACF zone.** Only 39 children aged 2 to 4 years (2·8%) were IGRA positive of 1,392 tested with a valid result which corresponds to an average ARTI of 0·9% (95% CI 0·6–1·2). The ARTI in children resident in the prior ACF was 0·6% (95% CI: 0·3% - 1·1%) and was 1·1% (95% CI: 0·7% - 1·5%) for the children resident in the 'no prior ACF' zone (Table 4, Fig 2). The distribution of the ARTI based on bootstrapping in the prior ACF is shifted to the left compared to the ARTI in the 'no prior ACF' zone. The ARTI was also lower in the 'prior ACF' compared to 'no prior ACF' in the IPW/MI model.

## Discussion

The estimated prevalence of microbiologically confirmed pulmonary tuberculosis in Karachi was 387 cases per 100,000 population (95% CI 276–498) with an estimated prevalence of 421

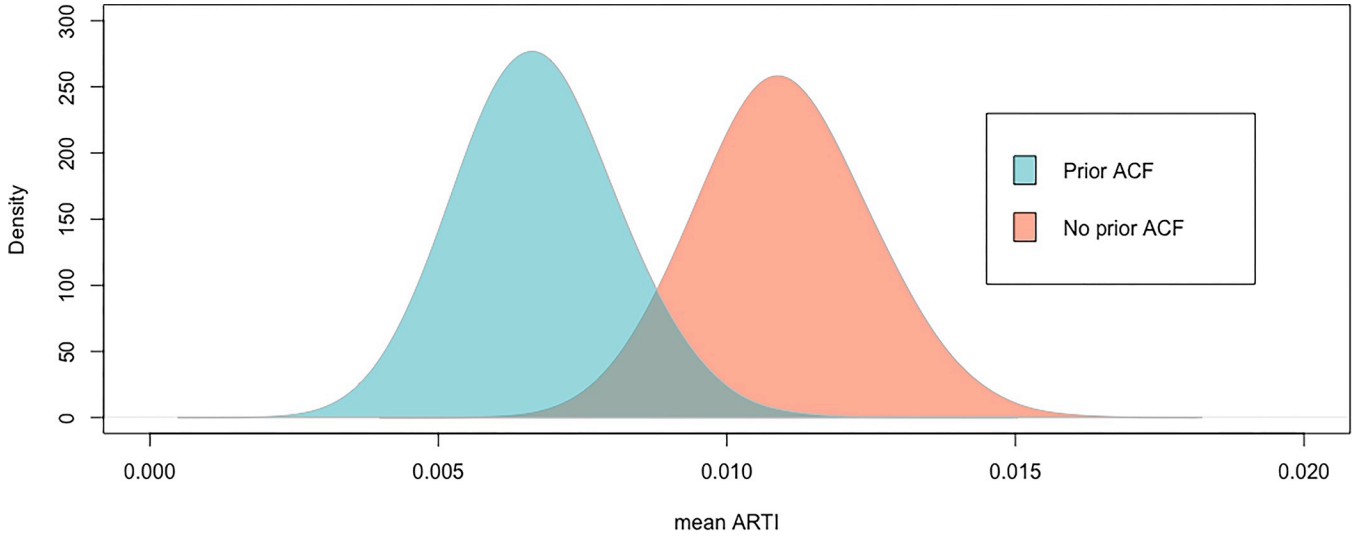

**Fig 2. Distribution of mean annual risk of *M.tuberculosis* infection (ARTI) by zone based on 20,000 bootstrap samples.**

cases [95% CI 276–567] per 100,000 in the 'no prior ACF' zone and 279 cases [95% CI 155–403] per 100,000 in the 'prior ACF'. The prevalence estimates were higher in men compared to women in the 'no prior ACF' zone irrespective of analytic method used and were similar in men compared to women in the 'prior ACF' zone. We also observed a lower ARTI in children aged 2 to 4 years in the prior ACF zone, 0.6% (95% CI 0·3–1·1) compared to 1·1% (95% CI 0·7–1·5) in the no prior ACF zone.

Our interpretation of these key findings are that they most likely reflect underlying important population differences of tuberculosis burden between zones which are historically contingent on causal processes [23]. A plausible explanation is that intensive ACF activities that had been ongoing in Korangi district for the preceding 7 years have decreased population tuberculosis prevalence and transmission through higher case detection, earlier diagnosis and reduced the duration of infectiousness. The similar burden observed in men compared to women in the 'prior ACF' zone is an interesting finding. Although no causal conclusions can be drawn from these data it is conceivable that more men than women participated in the prior ACF activities in Korangi district resulting in a differential impact on tuberculosis burden by gender. From our own experience during the pilot phase of the survey, we observed much lower participation in women which required a reconfiguration of the field teams to include more women field workers to be able to act as chaperones. This markedly increased female participation in our survey.

The sensitivity analysis which excluded all study participants with Xpert Ultra 'trace positive' only results found no difference between the zones suggesting that difference in tuberculosis prevalence is driven primarily by the difference in detection of subclinical paucibacillary disease. The lower burden of subclinical paucibacillary disease in the prior ACF zone compared to no prior ACF might have resulted from ongoing prior ACF activities which mopped up these cases, predominantly in the men. Interestingly, only one male study participant and 10 female study participants with Ultra 'trace positive' only results were identified in the prior ACF zone whilst 13 male study participants and 13 female study participants were identified in the no prior ACF zone.

Young children with *M. tuberculosis* infection act as sentinels of recent *M. tuberculosis* transmission [24], and interferon gamma release assays (IGRAs) are one way of inferring infection status and have a higher specificity compared to tuberculin skin tests (TST), especially in young recently BCG-vaccinated children [25]. Whilst acknowledging the differences in transmission dynamics between adults and children, choosing a younger age group for the IGRA survey had the advantage that the estimated ARTI reflects transmission within a relatively narrow period. In our study this was approximately the last 3 years, i.e. between the average birth date of the group and the date of the survey [21]. The trend towards a lower ARTI in the 'prior ACF' population compared to the 'no prior ACF' population irrespective of method used does suggest a reduced risk of transmission to the youngest in the population.

Approximately 70% of participants with microbiologically confirmed pulmonary tuberculosis reported no symptoms in our survey which was similar to the findings from the national Pakistan tuberculosis prevalence survey from 2010 and other prevalence surveys from the region [26,27]. Of note, only 26% of those who were found to have *M. tuberculosis* in their sputum actually started tuberculosis treatment. People with tuberculosis identified through our prevalence survey are likely to be reflective of cases found through untargeted ACF activities. It is as yet unclear if detecting and treating individuals found to have *M. tuberculosis* in their sputum following a positive CXR-screen, in whom symptoms have not become manifest or are not severe enough to warrant seeking healthcare, would result in a reduction of *M. tuberculosis* transmission at the population-level but it is conceivable [28]. However this would require identified individuals engaging in care and completing treatment. Whilst 16/40 (40%) of those

who were clinically assessed were deemed not to have tuberculosis requiring treatment, a significant proportion (32%) refused follow-up (41.5% in 'no prior ACF' versus 20.0% in 'prior ACF'). Clinical management of individuals found to have microbiologically confirmed pulmonary tuberculosis through CXR-based ACF who report no symptoms, especially those with Xpert Ultra 'trace positive' only (culture-negative) on sputum testing is complex [29], with clear guidance yet to be developed and currently at the discretion of tuberculosis programmes [30].

Strengths of the study include the innovative population-based random sampling strategy using GIS satellite technology implemented in an Asian urban mega-city. This study was also the first to use children under five years of age as part of a population based IGRA survey to provide a measure of recent *M. tuberculosis* transmission. We do however acknowledge several limitations. These include the suboptimal participation rates and sputum collection rates which were lower than the target of 85% and 90% respectively and differed by sex and zone in the adult survey [17]. Participation was also very low in the child IGRA survey and estimates maybe biased, especially if risk of IGRA-positivity is systematically different in those that participated versus those that did not. However, participation did not differ by ACF zone. We may have over-estimated the prevalence estimate as we defined pulmonary tuberculosis as any microbiological confirmation of *M. tuberculosis* including 'trace positive' only. More detailed information on coverage and intensity of ACF implementation activities with data on linkage to care and treatment completion by sex would have strengthened the argument for a population impact.

The tuberculosis field has a long tradition of conducting detailed epidemiological studies which supplement surveillance data but funding for these studies are increasingly difficult to secure. The findings of this paper highlight the value of designing, conducting and interpreting findings from a descriptive epidemiological study of tuberculosis burden which takes account of the historical context and interprets the findings through an eco-epidemiological lens [31]. Findings also highlight key research priorities such as the understanding the biological basis of Xpert Ultra 'trace positive' culture-negative disease [32] and whether paucibacillary (Xpert Ultra 'trace positive' only with no previous history of tuberculosis) subclinical disease should be included in prevalence estimates in the future. This study provides empirical population-level data which may be useful to mathematical modellers working on models forecasting the longer-term effects of ACF which will hopefully lead to better decision-making for global tuberculosis control.

## Supporting information

**S1 Checklist. STROBE statement—checklist of items that should be included in reports of observational studies.**
(DOCX)

**S2 Checklist.**
(DOCX)

**S1 Fig. Case notification rate of bacteriologically confirmed tuberculosis stratified by ACF zone from 2007 to 2017.**
(TIFF)

**S2 Fig. A. Study flowchart for adult pulmonary tuberculosis survey. B. Study flowchart for child *M. tuberculosis* infection survey.**
(TIF)

**S3 Fig. Map of Karachi illustrating variation in log population count and distribution of neighbourhood blocks sampled by zone.** A = whole Karachi. B = map zoomed in to show distribution of sampling units in the two zones (no prior ACF [Karachi South, Central and West] and prior ACF [Karongi district]). The shapefile used to generate the map was obtained from Humanitarian Data Exchange (https://data.humdata.org/dataset/cod-ab-pak; administrative level 2 boundary shapefile "pak_admbnda_adm2_wfp_20220909.shp"), which is shared under Creative Common Attributions 4.0 International (CC by License). The population data for 2017 for Karachi displayed in the maps was obtained from WorldPop Hub (https://hub.worldpop.org/geodata/summary?id=28175) which is also shared under the Creative Commons Attribution 4.0 International License.[22].
(TIFF)

**S4 Fig.**
(TIFF)

**S1 Text. Models estimating adult tuberculosis prevalence.**
(DOCX)

**S1 Table. Xpert Ultra 'trace positive' only case series (n = 38).**
(DOCX)

**S2 Table. Characteristics of participants and microbiological outcomes of the adult pulmonary tuberculosis survey stratified by district and sex.**
(DOCX)

**S3 Table. Sensitivity analysis excluding Xpert Ultra 'trace positive' culture negative cases.**
(DOCX)

**S4 Table. Adult pulmonary tuberculosis prevalence estimates stratified by zone and *katchi abadi* status (per 100,000 population).**
(DOCX)

## Acknowledgments

This study would not have been possible without the dedication, attention to detail and collaborative spirit of the entire survey team (field workers, counsellors, x-ray van drivers, phlebotomists and field supervisors).

## Author Contributions

**Conceptualization:** Palwasha Y. Khan, Chris Grundy, Falak Madhani, Maqboola Dojki, Liesl Page-Shipp, Nazia Khursheed, Waleed Rabbani, Najam Riaz, Saira Khowaja, Rabia Maniar, Uzma Khan, Ali Habib, James J. Lewis, Katharina Kranzer, Rashida A. Ferrand, Katherine L. Fielding, Aamir J. Khan.

**Data curation:** Palwasha Y. Khan, Falak Madhani, Saadia Saeed, Lamis Maniar, Maqboola Dojki, Liesl Page-Shipp, Nazia Khursheed, Waleed Rabbani, Najam Riaz, Saira Khowaja, Owais Hussain, Rabia Maniar, Ali Habib, Katherine L. Fielding.

**Formal analysis:** Palwasha Y. Khan, Falak Madhani, Saadia Saeed, Lamis Maniar, Owais Hussain, James J. Lewis, Katharina Kranzer, Rashida A. Ferrand, Katherine L. Fielding.

**Funding acquisition:** Liesl Page-Shipp, Najam Riaz, Saira Khowaja, Aamir J. Khan.

**Investigation:** Palwasha Y. Khan, Mohammed Shariq Paracha, Chris Grundy, Falak Madhani, Saadia Saeed, Lamis Maniar, Maqboola Dojki, Liesl Page-Shipp, Nazia Khursheed, Waleed Rabbani, Najam Riaz, Saira Khowaja, Owais Hussain, Rabia Maniar, Uzma Khan, Rashida A. Ferrand, Katherine L. Fielding.

**Methodology:** Palwasha Y. Khan, Mohammed Shariq Paracha, Chris Grundy, Falak Madhani, Saadia Saeed, Lamis Maniar, Maqboola Dojki, Liesl Page-Shipp, Nazia Khursheed, Saira Khowaja, Owais Hussain, Rabia Maniar, Uzma Khan, Sabira Tahseen, Ali Habib, James J. Lewis, Katharina Kranzer, Rashida A. Ferrand, Katherine L. Fielding.

**Project administration:** Palwasha Y. Khan, Mohammed Shariq Paracha, Falak Madhani, Saadia Saeed, Lamis Maniar, Maqboola Dojki, Nazia Khursheed, Najam Riaz, Saira Khowaja, Owais Hussain, Rabia Maniar, Katherine L. Fielding.

**Resources:** Falak Madhani, Lamis Maniar, Maqboola Dojki, Liesl Page-Shipp, Nazia Khursheed, Najam Riaz, Saira Khowaja, Rabia Maniar, Uzma Khan, Salman Khan, Syed S. H. Kazmi, Ali A. Dahri, Abdul Ghafoor, Sabira Tahseen, Ali Habib, Aamir J. Khan.

**Software:** Palwasha Y. Khan, Chris Grundy, Saadia Saeed, Lamis Maniar, Owais Hussain.

**Supervision:** Palwasha Y. Khan, Mohammed Shariq Paracha, Chris Grundy, Falak Madhani, Lamis Maniar, Maqboola Dojki, Liesl Page-Shipp, Saira Khowaja, Owais Hussain, Rabia Maniar, Uzma Khan, Salman Khan, Syed S. H. Kazmi, Ali A. Dahri, Sabira Tahseen, Ali Habib, Rashida A. Ferrand, Katherine L. Fielding.

**Validation:** Palwasha Y. Khan.

**Visualization:** Palwasha Y. Khan, Chris Grundy, Saadia Saeed, Lamis Maniar, Owais Hussain, Ali Habib.

**Writing – original draft:** Palwasha Y. Khan, Mohammed Shariq Paracha, Falak Madhani.

**Writing – review & editing:** Palwasha Y. Khan, Liesl Page-Shipp, Owais Hussain, Uzma Khan, Salman Khan, Syed S. H. Kazmi, Ali A. Dahri, Abdul Ghafoor, Sabira Tahseen, Ali Habib, James J. Lewis, Katharina Kranzer, Rashida A. Ferrand, Katherine L. Fielding, Aamir J. Khan.

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
