## [Decision Letter · Decision Letter 0]

9 Aug 2023

PGPH-D-23-00926

Insights into tuberculosis burden in Karachi, Pakistan: a concurrent adult tuberculosis prevalence and child M.tuberculosis infection survey

Dear Dr. Khan,

Thank you for submitting your manuscript to PLOS Global Public Health. After careful consideration, we feel that it has merit but does not fully meet PLOS Global Public Health’s publication criteria as it currently stands. Therefore, we invite you to submit a revised version of the manuscript that addresses the points raised during the review process.

Please pay attention to the reviewers comments when revising your manuscript. In particular, please review the case definition related to the inclusion of Xpert MTB/RIF Ultra TRACE only as cases. Further provide an update on the progress made related to the legality of sharing of data. 

We look forward to receiving your revised manuscript.

Kind regards,

Shaheed Vally Omar, Ph.D.

Academic Editor

Journal Requirements:

1) In the ethics statement in the Methods, you have specified that verbal consent was obtained. Please provide additional details regarding how this consent was documented and witnessed, and state whether this was approved by the IRB

2) Please include a complete copy of PLOS’ questionnaire on inclusivity in global research in your revised manuscript. Our policy for research in this area aims to improve transparency in the reporting of research performed outside of researchers’ own country or community. The policy applies to researchers who have travelled to a different country to conduct research, research with Indigenous populations or their lands, and research on cultural artefacts. The questionnaire can also be requested at the journal’s discretion for any other submissions, even if these conditions are not met.  Please find more information on the policy and a link to download a blank copy of the questionnaire here: https://journals.plos.org/globalpublichealth/s/best-practices-in-research-reporting. Please upload a completed version of your questionnaire as Supporting Information when you resubmit your manuscript.

2. Please provide separate figure files in .tif or .eps format only and remove any figures embedded in your manuscript file. Please also ensure all files are under our size limit of 10MB.

3. Some material included in your submission may be copyrighted. According to PLOS’s copyright policy, authors who use figures or other material (e.g., graphics, clipart, maps) from another author or copyright holder must demonstrate or obtain permission to publish this material under the Creative Commons Attribution 4.0 International (CC BY 4.0) License used by PLOS journals. Please closely review the details of PLOS’s copyright requirements here: PLOS Licenses and Copyright. If you need to request permissions from a copyright holder, you may use PLOS's Copyright Content Permission form.

Potential Copyright Issues:

Figure S3: please (a) provide a direct link to the base layer of the map (i.e., the country or region border shape) and ensure this is also included in the figure legend; and (b) provide a link to the terms of use / license information for the base layer image or shapefile. We cannot publish proprietary or copyrighted maps (e.g. Google Maps, Mapquest) and the terms of use for your map base layer must be compatible with our CC-BY 4.0 license. 

"

Additional Editor Comments (if provided):

Reviewers' comments:

Reviewer's Responses to Questions

**Comments to the Author**

1. Does this manuscript meet PLOS Global Public Health’s publication criteria? Is the manuscript technically sound, and do the data support the conclusions? The manuscript must describe methodologically and ethically rigorous research with conclusions that are appropriately drawn based on the data presented.

Reviewer #1: Yes

Reviewer #2: Partly

2. Has the statistical analysis been performed appropriately and rigorously?

Reviewer #1: Yes

Reviewer #2: No

3. Have the authors made all data underlying the findings in their manuscript fully available (please refer to the Data Availability Statement at the start of the manuscript PDF file)?

Reviewer #1: No

Reviewer #2: No

4. Is the manuscript presented in an intelligible fashion and written in standard English?

Reviewer #1: Yes

Reviewer #2: Yes

5. Review Comments to the Author

Reviewer #1: Review PGPH-D-23-00926: Insights into tuberculosis burden in Karachi, Pakistan: a concurrent adult tuberculosis prevalence and child M.tuberculosis infection survey

The authors present both a study of the prevalence of pulmonary tuberculosis in adults, who they define as ≥15 years of age, and a study on annual risk of M. tuberculosis infection (ARTI) in children 2-4 years of age from the same communities. This is an interesting concept, and the results are important, showing a trend towards lower TB prevalence and ARTI in the area where active TB case finding (ACF) has been ongoing compared to areas where this was not routinely done. The reviewer does not have many comments, as the manuscript is well written and the study well pergormed.

Comments:

1. In the title “M.tuberculosis” should be “Mycobacterium tuberculosis”. In the manuscript (MS) the authors abbreviate “Mycobacterium tuberculosis” as “M.tuberculosis” without a space, but thereafter inconsistently add or delete the space – the correct abbreviation is “M. tuberculosis” (with space between M. and tuberculosis)

2. Abstract: Suggest add: “Pakistan” after “Karachi”

3. Abstract and MS: Spelling of “bacteriologically-confirmed” pulmonary tuberculosis – inconsistent in abstract and MS with/without hyphen, but correct spelling is without a hyphen in this wording even though it is followed by a noun

4. Methods, Study design and setting, last line: Should use “ACF” as already previously abbreviated in MS

5. Methods, Participants, line 9: Suggest state that adults are for this study defined as ≥ 15 years of age: “All adults, defined for this study as aged ≥15 years who were resident…”. The question is whether in Pakistan the 15-<18-year-old participants can give their own consent for study participation or whether their parents/legal guardians should give consent, even though verbal as in this study?

6. Study procedures, line 14: Suggest add “cultures” after “liquid and solid media”

7. Child M. tuberculosis infection survey using interferon-gamma release assays (IGRA), line 9: suggest “a 6-month course…”

8. Page 6, under “Statistical methods” there is a subheading “Statistical analysis” followed by a paragraph with subheading “Child survey case definition”. The reviewer thinks that, because the paragraph following “Child survey case definition” is again about the adults, it would be much better that the “Child survey case definition” moves one paragraph further down t keep adults and children’ data analyses together.

9. Page 6, second to last paragraph, last line: The missing IGRA data in children “(n=1,119)” falls somewhere out of the blue – should this not be mentioned in the results?

10. Results, Adults: I understand that the authors had a specific definition for “microbiologically confirmed pulmonary tuberculosis”, but does CAD4TB clearly distinguish between old scarring of TB and “active” PTB, and could other conditions be similar to TB on CAD4TB? It seems that most adults diagnosed with PTB either did not have PTB or did not consider themselves as having PTB needing treatment – could this be an overestimation due to “M. tuberculosis infection” or previous PTB rather than “active” TB disease? This is somewhat difficult to interpret in this study – how do these “cases” that seem not to have “disease” effect the two zones (prior and non-prior ACF), if left out?

11. Results: A concern regarding the prevalence of infection amongst the children is the very low inclusion/interpretable IGRA result “rate” – only 1392/2511 (55.4%) – do the authors think that this could be an important bias in the study? What were the main reasons for refusal of participation? Interesting that contact with TB in the past two years was higher in the prior ACF group and IGRA-positivity lower – comment?

12. Discussion, page 10, 4th paragraph, line 3: “Of note, only 20% of those who were…” – this is not the same as in the results section which indicates 26% - which is correct?

13. Finally, there is no mention of HIV positivity – is this an issue at all in Karachi?

Reviewer #2: In this manuscript, Khan et al. present the findings from concurrent adult TB prevalence and paediatric TB infection surveys conducted in 4 districts of Karachi, Pakistan, in 2018-2019. They compare the results from the surveys in one district which had previously been exposed to active case-finding (ACF) interventions to those from three other districts which had had no, or limited, exposure to previous active case-finding interventions (non-ACF districts). They report trends of both lower adult TB prevalence and lower paediatric annual risk of TB infection (ARTI) in the ACF exposed district compared to the non-ACF exposed districts. While being cautious about drawing causal inferences, the authors conclude that one plausible explanation for the observed differences is that prior ACF activities had decreased TB prevalence and transmission in the ACF district. While it is an important study, there are two major issues that need to be addressed before it can be accepted for publication.

Major issues

1. The inclusion of Xpert MTB/RIF Ultra trace only as cases in the adult prevalence survey case definition is problematic.

A total of 93 probable and definite bacteriologically confirmed pulmonary tuberculosis cases were included in the analysis. Of these 93 ‘cases’, 40% (37/93) were included on the basis of only an Xpert Ultra trace result. While no participants with Xpert Ultra trace were culture positive, the number that were culture negative is not reported. The decreased specificity of the Xpert Ultra test compared to the Xpert MTB/RIF has been well described, particularly if trace results are regarded as positive, and in patients with a recent prior history of TB.

The proportion of cases that were Xpert Ultra trace only was substantially higher in the non-ACF ‘cases’ (26/53, 49%) compared to the ACF (11/40, 27.5%). Excluding these Ultra trace only results from the case definition would almost entirely remove the observed difference in the adult TB prevalence between the ACF and non-ACF areas. The Ultra trace only ‘cases’ therefore drive the reported difference in the adult pulmonary TB prevalence estimates, one of the central findings of the study.

Furthermore, amongst the 40 ‘cases’ that were clinically assessed, TB was excluded in 16 participants with 12 of these 16 having trace only results. Once again calling into question the appropriateness of including Ultra trace only in the case definition.

Lastly, the inclusion of the Ultra trace only participants in the case definition is not in keeping with the primary or secondary analysis outlined in the published protocol. The protocol includes the following case definitions for the adult pulmonary TB prevalence survey component:

a. “Definite bacteriologically confirmed pulmonary tuberculosis case is defined as study participant with one culture positive sputum specimen and at least one of the following conditions: Xpert Ultra-positive [or] Culture-positive (either LJ or MGIT) in another specimen.”

b. “Probable bacteriologically confirmed pulmonary tuberculosis case is defined as a study participant with an Xpert Ultra-positive sputum specimen but culture-negative on one or both sputum samples or a study participant with a single culture- positive sputum where laboratory cross-contamination has been excluded.”

The protocol notes that “a sensitivity analysis using the ‘probable TB case’ definition will also be undertaken.” However, neither the definite nor the probable definitions include Ultra trace only in the definition.

Recommendation: The authors revise the case definition to be in keeping with the probable case definition in the protocol and excluding the trace only ‘cases’.

2. Selection of the three non-ACF districts: While the rationale for the selection of the prior ACF district is well described in the manuscript, it appears that the 3 no prior ACF districts were chosen on the basis of convenience. Substantial spatial heterogeneity in TB burden has been well described in many cities, and thus is an alternative plausible explanation for the differences. The higher average population within 100m in the non prior ACF zone could potentially account for some of the difference in the paediatric IGRA prevalence and annual risk of infection. While the number of adults in neighbourhood blocks was included in the multiple imputation of missing IGRA results, indicating that the authors identified this as an important factor, it does not appear that the estimates were adjusted for district level population differences.

Recommendation: Authors should consider including district (or preferably neighbourhood block) characteristics in the analysis of the ARTI, and/or include discussion of alternative explanations.

Minor issue

3. Providing a supplementary table similar to table 1 with the data disaggregated by the 4 districts (as opposed to providing aggregated data for the 3 non ACF districts) would assist the reader to assess whether there was substantial heterogeneity across the 3 non ACF districts.

6. PLOS authors have the option to publish the peer review history of their article (what does this mean?). If published, this will include your full peer review and any attached files.

**Do you want your identity to be public for this peer review?** For information about this choice, including consent withdrawal, please see our Privacy Policy.

Reviewer #1: No

Reviewer #2: No

---

## [Decision Letter · Decision Letter 1]

24 Apr 2024

PGPH-D-23-00926R1

Insights into tuberculosis burden in Karachi, Pakistan: a concurrent adult tuberculosis prevalence and child M.tuberculosis infection survey

Dear Dr. Khan,

Thank you for submitting your manuscript to PLOS Global Public Health. After careful consideration, we feel that it has merit but does not fully meet PLOS Global Public Health’s publication criteria as it currently stands. Therefore, we invite you to submit a revised version of the manuscript that addresses the points raised during the review process.

Specifically, the reviewer highlights some errors in the text and provides suggestions for clarity and correctness. Please read the manuscript carefully to identify additional areas where revisions may be required which the reviewer has not specified.

We look forward to receiving your revised manuscript.

Kind regards,

Jennifer Tucker, PhD

Staff Editor

Journal Requirements:

2. We do not publish any copyright or trademark symbols that usually accompany proprietary names, eg  ©, ®, ™  (e.g. next to drug or reagent names). Please remove all instances of trademark/copyright symbols throughout the text, including  ™ & ® on pages 4 &5.

Additional Editor Comments (if provided):

Reviewers' comments:

Reviewer's Responses to Questions

**Comments to the Author**

1. If the authors have adequately addressed your comments raised in a previous round of review and you feel that this manuscript is now acceptable for publication, you may indicate that here to bypass the “Comments to the Author” section, enter your conflict of interest statement in the “Confidential to Editor” section, and submit your "Accept" recommendation.

Reviewer #1: All comments have been addressed

2. Does this manuscript meet PLOS Global Public Health’s publication criteria? Is the manuscript technically sound, and do the data support the conclusions? The manuscript must describe methodologically and ethically rigorous research with conclusions that are appropriately drawn based on the data presented.

Reviewer #1: Yes

3. Has the statistical analysis been performed appropriately and rigorously?

Reviewer #1: Yes

4. Have the authors made all data underlying the findings in their manuscript fully available (please refer to the Data Availability Statement at the start of the manuscript PDF file)?

Reviewer #1: Yes

5. Is the manuscript presented in an intelligible fashion and written in standard English?

Reviewer #1: Yes

6. Review Comments to the Author

Reviewer #1: The authors have addressed the comments from the first review. The reviewer has only minor suggested corrections/comments.

Minor comments:

Introduction

- 2nd paragraph, line 4: Suggest delete “etc.” – not very scientific and not needed, as the examples start with “e.g.,”

- 2nd last line page 3: “examined” should be “examine”

Study design and setting

- line 4: “with had” should be “which had”

Child M. tuberculosis infection… (page 5)

- line 3: Suggest “…within the last…”

Page 7

- suggest sub-headings “Ethics approval” and “Role … source” to have similar font size/bold than “Statistical methods”

Table 1

- Suggest change “Known TB contact with last 2 years” to “Known TB contact within last 2 years”

Discussion

- Page 10, 3rd paragraph, line 1: Suggest use “…Xpert Ultra ‘trace’ only results…” (add “Xpert” and “results”)

- Page 10, 4th paragraph, last line: Should “the population” not be either “this population" or "the ‘prior ACF’ population?

- Page 11, 2nd paragraph, line 8: I think “then” at the end of the sentence should be deleted.

References

- Ref 3 – why all uppercase for author and title?

- Ref 10 – Journal name abbreviation incorrect (double?)

- Ref 17 – URL and when accessed?

7. PLOS authors have the option to publish the peer review history of their article (what does this mean?). If published, this will include your full peer review and any attached files.

**Do you want your identity to be public for this peer review?** For information about this choice, including consent withdrawal, please see our Privacy Policy.

Reviewer #1: No

---

## [Decision Letter · Decision Letter 2]

28 May 2024

Insights into tuberculosis burden in Karachi, Pakistan: a concurrent adult tuberculosis prevalence and child M.tuberculosis infection survey

PGPH-D-23-00926R2

Dear Dr. Khan,

We are pleased to inform you that your manuscript 'Insights into tuberculosis burden in Karachi, Pakistan: a concurrent adult tuberculosis prevalence and child M.tuberculosis infection survey' has been provisionally accepted for publication in PLOS Global Public Health. There are a couple of minor grammatical suggestions from Reviewer #1 that I would like you to review. 

Best regards,

Leonardo Martinez

Academic Editor

Reviewer Comments (if any, and for reference):

Reviewer's Responses to Questions

**Comments to the Author**

1. If the authors have adequately addressed your comments raised in a previous round of review and you feel that this manuscript is now acceptable for publication, you may indicate that here to bypass the “Comments to the Author” section, enter your conflict of interest statement in the “Confidential to Editor” section, and submit your "Accept" recommendation.

Reviewer #1: All comments have been addressed

2. Does this manuscript meet PLOS Global Public Health’s publication criteria? Is the manuscript technically sound, and do the data support the conclusions? The manuscript must describe methodologically and ethically rigorous research with conclusions that are appropriately drawn based on the data presented.

Reviewer #1: Yes

3. Has the statistical analysis been performed appropriately and rigorously?

Reviewer #1: Yes

4. Have the authors made all data underlying the findings in their manuscript fully available (please refer to the Data Availability Statement at the start of the manuscript PDF file)?

Reviewer #1: Yes

5. Is the manuscript presented in an intelligible fashion and written in standard English?

Reviewer #1: Yes

6. Review Comments to the Author

Reviewer #1: Review:

The manuscript is much improved.

Three minor suggested corrections and 1 comment:

Page 3, third paragraph, line 2: add "to" to read: "...by tuberculosis to care early..."

Page 4, 2nd paragraph, line 3: “scaled-up” (hyphen, as is followed by a noun)

Page 7, 1st line “ethics review committees…

Page 10, Discussion, first paragraph. The estimated prevalences presented here are different from those presented in the last paragraph on page 8 – it may confuse the reader, so maybe just refer the reader to Table 2, as this will clarify this point.

7. PLOS authors have the option to publish the peer review history of their article (what does this mean?). If published, this will include your full peer review and any attached files.

**Do you want your identity to be public for this peer review?** For information about this choice, including consent withdrawal, please see our Privacy Policy.

Reviewer #1: No
